# DreamEdit: Subject-driven Image Editing

♠,♣Tianle Li, ♠Max Ku*, ♠,♣Cong Wei*, ♠,♣Wenhu Chen
♠University of Waterloo
♣Vector Institute, Toronto
{t29li,max.ku,cong.wei,wenhuchen}@uwaterloo.ca

Reviewed on OpenReview: https://openreview.net/forum?id=P9haooN9v2

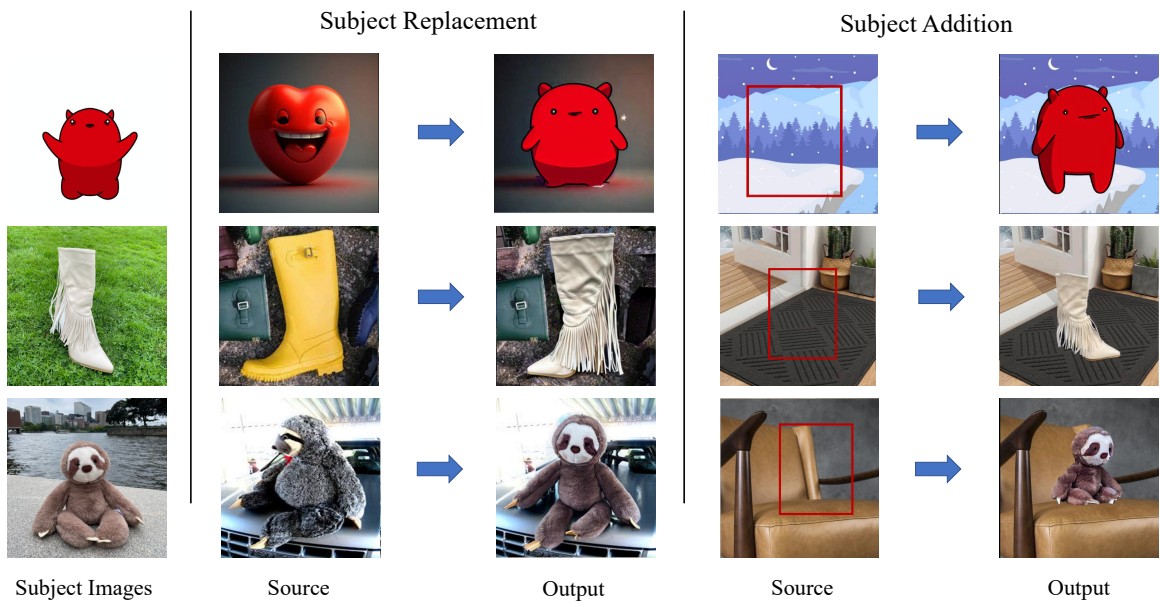

Figure 1: The leftmost column is the customized subject, the middle column is the Subject Replacement task, the rightmost column is the Subject Addition task. The output is the generated results by `DreamEditor`

## Abstract

Subject-driven image generation aims at generating images containing customized subjects, which has recently drawn enormous attention from the research community. Nevertheless, the previous works cannot precisely control the background and position of the target subject. In this work, we aspire to fill the void of the existing subject-driven generation tasks. To this end, we propose two novel subject-driven editing sub-tasks, i.e., Subject Replacement and Subject Addition. The new tasks are challenging in multiple aspects: replacing a subject with a customized one can totally change its shape, texture, and color, while adding a target subject to a designated position in a provided scene necessitates a rational context-aware posture of the subject. To conquer these two novel tasks, we first manually curate a new dataset called `DreamEditBench` containing 22 different types of subjects, and 440 source images, which cover diverse scenarios with different difficulty levels. We plan to host `DreamEditBench` and hire trained evaluators for enable standardized human evaluation. We also devise an innovative method `DreamEditor` to resolve these tasks by performing iterative generation, which enables a smooth adaptation to the customized subject. In this project, we conduct automatic and human evaluations to understand the performance of our `DreamEditor` and baselines on `DreamEditBench`. We found that the new tasks are challeng-

ing for the existing models. For Subject Replacement, we found that the existing models are particularly sensitive to the shape and color of the original subject. When the original subject and the customized subject are highly different, the model failure rate will dramatically increase. For Subject Addition, we found that the existing models cannot easily blend the customized subjects into the background smoothly, which causes noticeable artifacts in the generated image. We believe `DreamEditBench` can become a standardized platform to enable future investigations towards building more controllable subject-driven image editing. Our project and benchmark homepage is `https://dreameditbenchteam.github.io/`.

## 1 Introduction

Can you imagine your favorite pet cat going to any place that any other cat has been to, just by picking a photo for them? Can you picture your most familiar objects appearing in authentic locations around the world, simply by providing a photograph of the surroundings? The replacement or incorporation of personalized subjects into a source image can present an exhilarating challenge, and these possibilities remain largely unexplored in previous endeavors.

Current subject-driven image generation (as proposed by Ruiz et al. (2023); Gal et al. (2022); Chen et al. (2023b)) primarily relies on textual prompts to synthesize images with the given subject situated in diverse visual scenes. However, text-guided subject-driven image generation struggles to control the location or pose of the subject, background scene, image layout, etc. This lack of controllability hinders its applicability in real-world applications. In contrast, text-guided image editing (Dong et al., 2017; Li et al., 2020b;a) can precisely control the subject location, image layout, and background scene but falls short in controlling the synthesized subjects. Therefore, we are motivated to close the gap between subject-driven generation and image editing, proposing the new task of subject-driven image editing.

In this work, we propose two novel subject-driven image editing subtasks, i.e., Subject Replacement and Subject Addition as shown in Figure 1. The goal of subject replacement is to replace a subject from a source image with a customized subject. In contrast, the aim of the subject addition task is to add a customized subject to a desired position in the source image. These two tasks pose challenges in the following aspects: 1) the generated subject needs to maintain a similar location and pose to the source image, and 2) the generated subject should blend in with the environment realistically.

To evaluate the newly proposed task of subject-driven image editing, we first manually curate a new dataset, `DreamEditBench`. The dataset consists of examples with (source image, target subject) as inputs, and the model needs to generate a target image with the target subject appearing in the source image. The target subject is represented by a few reference images. The dataset spans more than 20 subject classes and 440 carefully picked source images with diverse backgrounds. For the addition task, we manually labeled different proper bounding boxes in the source image to specify the desired location for the target subject.

Further, we develop a novel iterative generation method, i.e., `DreamEditor`. Our model takes three steps: (1) we fine-tune the text-to-image generation model with DreamBooth (Ruiz et al., 2023) on the target subject images to associate the subject with a special token [V]. (2) we adopt off-the-shelf segmentation tool to locate the region for inpainting and then use DDIM inversion to re-paint the region with the target subject, which is guided by the special token [V]. (3) However, we observe that the generated subject can still retain unwanted properties from the source subject. Therefore, we adopt an iterative in-painting, where we feed the generated result as input to perform another round of in-painting. After 2-3 iterations, we can observe the re-painted subject will gradually match the target subject while fitting naturally with the background. We compare `DreamEditor` with several baselines with human evaluation and show that `DreamEditor` can achieve better overall scores consistently on both tasks.

After obtaining human evaluation results for `DreamEditor` and baselines, we observe that it is easier for the model to replace or add the target to the source image in some cases than others. For Subject Replacement, the source subject can be seamlessly replaced by the target one when they share similar features like colors and shape, thus merely one iteration suffices to yield satisfactory output. However, when it comes to the

scenario that the target subject differs dramatically from the source, it will entail longer iterations to fit to the target gradually as shown in Figure 2. Similarly, for the subject addition task, the position and posture of the added target are more strictly stipulated for some of the backgrounds. As shown in the right column in Figure 1, the generated sloth plush should be precisely positioned on the chair rather than hovering in the air. For these cases, we note that `DreamEditor` can generate an irrational posture or position for the target subject in the initial iteration, but rectify it to a rational one in later iterations. Accordingly, we further divide the `DreamEditBench` into easy and hard levels. The sharp drop experiment results from easy to hard subsets reflect that the applied methods do struggle with the hard source images more than the easy ones. Moreover, we notice that the human evaluation results diverge largely from the automatic evaluation results. None of the automatic metrics like CLIP (Radford et al., 2021) and DINO (Caron et al., 2021) is able to reflect the success rate. Therefore, we advocate the necessity of conducting a rigorous human evaluation and plan to build a platform for more rigorous evaluation.

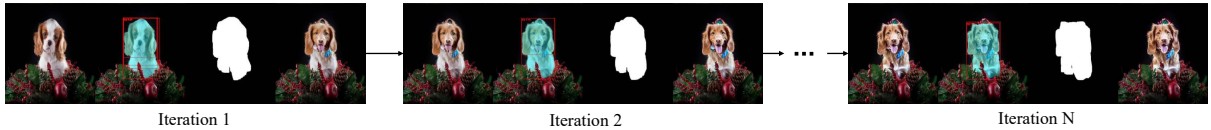

Figure 2: The visualization of `DreamEditor` to iteratively refine the generated target subject.

In a nutshell, the major contributions of this work are as follows:

**1. Task Definition.** We propose two novel new subject-driven image editing tasks, i.e. subject replacement and subject addition, and identify their challenges.

**2. Benchmark.** To standardize the evaluation of the two tasks, we collect the `DreamEditBench`, containing same-typed subjects and background images.

**3. Method.** We develop `DreamEditor`, a novel iterative generation method, to enable gradual adaptation from the source subject to the target one.

## 2 Related Work

**Conditional Image Editing with Deep Generative Models.** The concept of Image-to-Image translation was introduced where a model is trained to learn a mapping between images distributions from two different domains so that an image from one domain can be translated to another domain (Isola et al., 2017; Zhu et al., 2017a;b; Huang et al., 2018; Park et al., 2020). Image-to-Image translation models are widely used in style transfer and photo enhancement tasks. However, Image-to-Image translation models lack the ability to edit locally on an image. Researchers have found that the generative prior can be used to manipulate parts of the generated image. To edit real images, they first need to be inverted into the latent space. This concept, known as 'Inversion', allows for detailed image editing. The idea of GAN inversion was first introduced by the method of Zhu et al. (2016), and a significant effort has been made on efficient GAN inversion (Perarnau et al., 2016; Creswell & Bharath, 2019; Abdal et al., 2019) and applying GAN inversion to image editing applications (Bau et al., 2019; Patashnik et al., 2021; Tov et al., 2021).

Recently, diffusion models (Dhariwal & Nichol, 2021) dominate the field of image generation due to their superiority in generating realistic images. A denoising diffusion model learns to gradually denoise data starting from pure noise and a UNet (Ronneberger et al., 2015) is leveraged as the network architecture. The objective of a diffusion model is to learn a reverse diffusion process that restores structure in data to synthesize realistic data (Ho et al., 2020). Text-to-Image Diffusion Models are later proposed and achieve state-of-the-art image generation (Nichol et al., 2022; Ramesh et al., 2022; Saharia et al., 2022). In these methods, a pre-trained text encoder such as the CLIP (Contrastive Language–Image Pre-training) (Radford et al., 2021) is used to bridge the gap between textual descriptions and visual content. Diffusion-based image editing methods were also introduced in Lugmayr et al. (2022); Hertz et al. (2022); Mokady et al. (2022); Parmar et al. (2023); Couairon et al. (2022); Kawar et al. (2022).

**Subject-Driven Image Editing with Personalization of Pretrained Diffusion Model.** As previously mentioned, image editing methods heavily rely on latent prior. In subject-driven editing tasks, the subject is

often out of the latent distribution and thus it would be difficult to maintain subject fidelity when performing image editing. One way is to assume that a large-scale text-to-image diffusion model already understands a partial concept of the subject and that the subject can be derived from the right input text vector. Textual Inversion is done by performing a gradient update to optimize a text token vector to such that the text token vector represents the subject with a tiny portion of subject data (Gal et al., 2022). On the other hand, the straightforward approach is to fine-tune the large-scale text-to-image diffusion model to learn the concept of the subject. DreamBooth performs efficient fine-tuning on a large-scale diffusion model with a tiny portion of subject data (Ruiz et al., 2023). Following subject personalization works (Chen et al., 2023a; Shi et al., 2023; Tewel et al., 2023) investigate more efficient ways to perform subject-driven image generation. More recently, there are concurrent works like BLIP-Diffusion (Li et al., 2023a), CustomEdit (Choi et al., 2023) and PhotoSwap (Gu et al., 2023) working on the task of subject swapping. Our work differs from these works in two aspects: (1) the previous work only focuses on subject replacement, (2) our method can work on cases where the source and target subjects differ significantly. Our iterative method can modify the appearance iteratively to increase subject fidelity. In the context of subject addition, the objectives and approaches significantly deviate from those employed in image composition (Niu et al., 2021; Zhang et al., 2021). Contrary to the mere placement and subsequent harmonization typical in image composition, subject addition is inherently a generative process. This process not only introduces the subject into the background but also dynamically adjusts various attributes of the subject, such as posture, expression, and viewing angle, to achieve seamless integration with the existing background. This adaptation is executed automatically with diffusion process, underscoring the advanced capabilities of this approach in generating contextually coherent images.

## 3 Preliminary

**Text-to-Image Diffusion Models.** A diffusion probabilistic model is a latent variable model that is trained to learn the image distribution by reversing the diffusion Markov chain. Specifically, we are interested in large text-to-image diffusion models pre-trained on large-scale text-image pairs (Rombach et al., 2021). The model consists of a CLIP (Radford et al., 2021) text encoder $\Gamma$, and a U-Net (Ronneberger et al., 2015) based conditional diffusion model $\epsilon_\theta$. Given a text prompt $Q$, the text encoder $\Gamma$ generates a conditioning vector $\Gamma(Q)$. With a randomly sampled noise $\epsilon \sim \mathcal{N}(0, I)$ and the time step t, we can get a noised image or latent code $z_t = \alpha_t \mathbf{x} + \sigma_t \epsilon$ where $\mathbf{x}$ is the input image, $\alpha_t$ and $\sigma_t$ are the coefficients that control the noise schedule. Then the conditional diffusion model $\epsilon$ is trained with the denoising objective:

$$\mathbb{E}_{x,Q,\epsilon,t}[\|\epsilon - \epsilon_\theta(z_t, t, \Gamma(Q))\|_2^2] \tag{1}$$

Where $\epsilon$ is trained to predict the noise condition on the noisy latent $z_t$, a text prompt $Q$, and the time step $t$. At inference, given a noise latent $z_T$, the noise is gradually removed by sequentially predicting it using $\epsilon_\theta$ for $T$ steps. A more detailed description is given in the supplementary material.

**Subject-Driven Text-to-Image Generation.** Subject-driven generation models (Ruiz et al., 2023) often fine-tune a pre-trained text-to-image generation model $\epsilon_\theta$ on a small set of demonstrations $\mathbb{C}_s$ of subject $s$, which contains a set of image and text description pairs $\mathbb{C}_s = \{(\mathbf{x_i}, Q_i)\}_{i=1}^K$, where $x_i$ means the $i^{th}$ image of subject $s$, while text description $Q_i$ of image $x_i$ contains a special text token $V_s$ that is bound to the subject $s$ (Ruiz et al., 2023). To customize a diffusion model $\epsilon_{\theta_s}$, we optimize the objective:

$$\mathbb{E}_{(x,Q)\sim\mathbb{C}_s,\epsilon,t}[\|\epsilon - \epsilon_{\theta_s}(z_t, t, \Gamma(Q))\|_2^2] \tag{2}$$

Where $\epsilon_{\theta_s}$ is trained to always denoise for the images of the subject $s$ when condition on a text description that contains the special text token $V_s$.

**DDIM Inversion for Real Image Edition.** Text-guided editing of a real image with generative models often requires inverting the given image and textural prompt. Essentially, this means identifying an initial noise vector that can generate the given real image, while maintaining the model's ability to edit. We

leverage the DDIM inversion scheme (Dhariwal & Nichol, 2021; Song et al., 2021) to encode a given image $z_0 = x_0$ onto a latent variable $z_T$.

$$z_{t+1} = \sqrt{\frac{\alpha_{t+1}}{\alpha_t}} z_t + \left( \sqrt{\frac{1}{\alpha_{t+1}} - 1} - \sqrt{\frac{1}{\alpha_t} - 1} \right) \cdot \epsilon_\theta(z_t, t) \tag{3}$$

Based on the assumption that the ODE process can be revered in the limit of small steps, $z_T$ is computed by reversing the deterministic DDIM sampling. $\epsilon_\theta(z_t, t)$ is an unconditional model or equivalently a conditioning model with text $\emptyset$. i.e. $\epsilon_\theta(z_t, t, \emptyset)$.

## 4 Benchmark

### 4.1 Problem Definition

**Subject Replacement** The Subject Replacement task aims to generate a new image by replacing the subject in a source image with the subject from a set of target subject images, guided by a general description. Formally, given the target subject images set $\mathbb{C}_s$, a text prompt $Q^*$ describing the replaced image, and an image containing the same-typed subject to be replaced $S_{rep}$, the task of Subject Replacement is targeted at obtaining a transformed image $R = \Omega(\mathbb{C}_s, Q^*, S_{rep})$, so that the feature of the replaced subject in $R$ aligns with the target subject in $\mathbb{C}_s$, and simultaneously, the background in $S_{rep}$ is still preserved after transformation. As shown in the first row of Figure 1, we replace the heart cartoon in the source image $S_{rep}$ with the red evil cartoon in the target subject images $\mathbb{C}_s$ and preserve the background information in $S_{rep}$.

**Subject Addition** The Subject Addition task seeks to place the target subject at a designated position in a background image. Similar to the Subject Replacement task, given a set of images of the target subjects $\mathbb{C}_s$, a text prompt $Q^*$ describing the after-placement image, a suitable background image $S_{add}$ for the customized subject, and a manually labeled bounding box $Bbox$ specifying the addition position, the task can be formulated formally as $A = \Upsilon(\mathbb{C}_s, Q^*, S_{add}, Bbox)$. In the Subject Addition task, it is crucial to ensure the rationality of the interaction between the generated subject and the context. For instance, the grey sloth plush in the last column of Figure 1 should always be placed on the chair other than hovering in the air. In addition, the background details are necessitated to be preserved.

### 4.2 Data Collection

To standardize the evaluation of the two proposed tasks, we curate a new benchmark, i.e. `DreamEditBench`, consisting of 22 subjects in alignment with DreamBooth [Ruiz et al. (2023)] with 20 images for each subject correspondingly. For the subject replacement task, we collect 10 images for each type, which include same-typed source subjects in diverse environments as shown in Figure 6a. The images are retrieved from the internet with the search query "a photo of [Class name]", and the source subject should be the main subject in the image which dominates a major part of the photo. For the subject addition task, we collect 10 reasonable backgrounds for each type of subject as shown in Figure 6b. In the meantime, we manually designate the specific location the target subject should be placed with a bounding box in the background. To collect the specific backgrounds for each subject, we first brainstorm and list the possible common environments of the subjects, then we search the listed keywords from the internet to retrieve and pick the backgrounds.

### 4.3 Dataset Analysis

We notice that for Subject Replacement, some of the collected source images share similar features with the target, while others differ dramatically from the target in color, texture, or shape. According to our observation, the significant distinction in features can result in the degradation of model performance. For instance, it is easier to adapt to the target teapot for the source in the left column than the right in Figure 6a, as the left teapots share a more similar color to the target. Therefore, we divide the collected source images into 150 easy ones and 70 hard ones. For Subject Addition, the model is likely to generate the subject more reasonably in some of the backgrounds than the others. For instance, if the view and style of

the background images diverge far away from the provided set of subjects, it is hard for the stable diffusion model to synthesize a target subject adapting to the unseen situation (e.g. the right column of a candle in Figure 6b is classified to hard as it requires the generation of the top view). Moreover, another example is the bear plush in Figure 6b, the bottom of the generated bear plush is strictly mandated to be at the bottom edge of the bounding box, otherwise, it makes no sense to place a plush in the air or below the bay window. Accordingly, we divide the 220 backgrounds into 122 easy-typed images and 98 hard ones.

## 5 Methodology

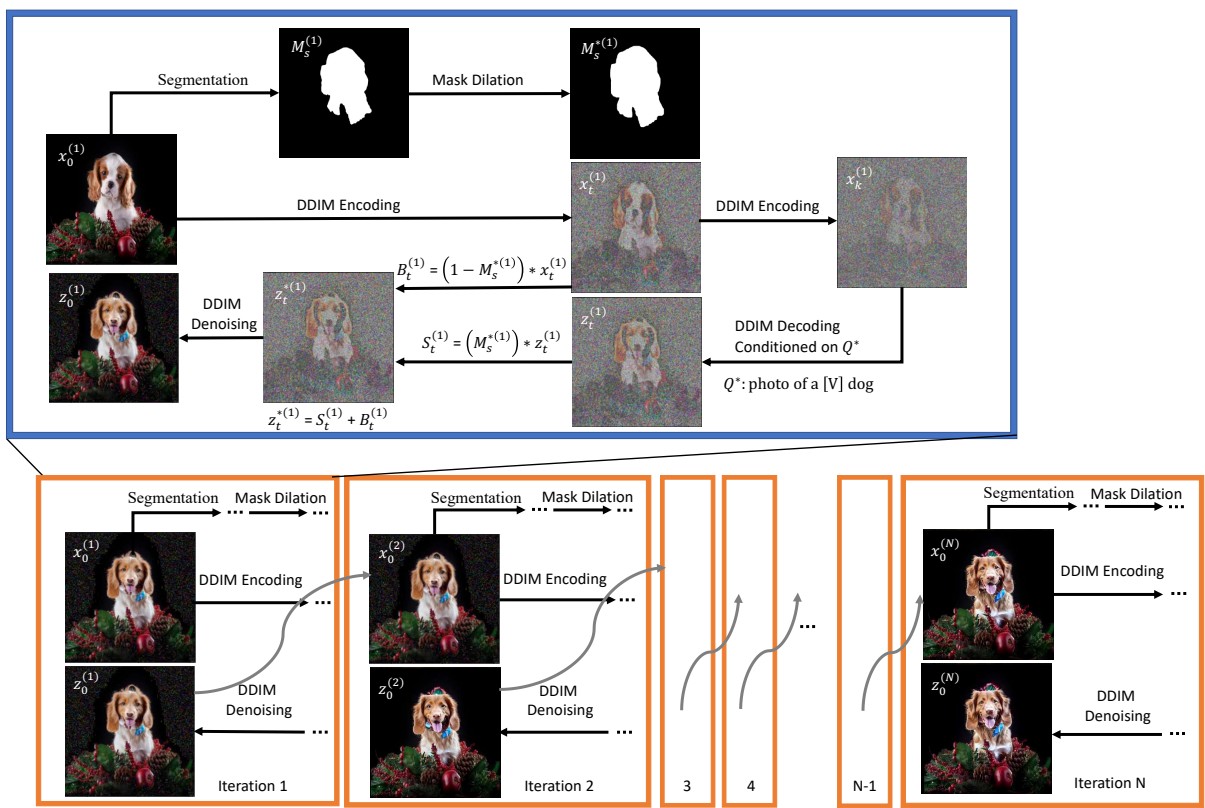

Figure 3: `DreamEditor` generates the results iteratively. The output of the last iteration will serve as the input for the next. In each iteration, it leverages the dilated mask of subject segmentation and the specialized prompt to guide the DDIM inversion and gradually in-paint a subject more similar to the target one.

To effectively resolve the two tasks, we build a novel system `DreamEditor` as an iterative adapter to the customized subject. `DreamEditor` can realize customized subject replacement or addition with the following steps: 1) Given a set of the target subject images, we fine-tune the text-to-image model with DreamBooth Ruiz et al. (2023) to associate the target subject to a special token. 2) We initialize the source image with two different strategies. 3) We leverage DDIM inversion guided by the segmentation mask and target text prompt to generate the initial result. 4) We repeat the process iteratively, where the output will become the input for the next round. An overview of the proposed `DreamEditor` is shown in Figure 3.

**Initialization** Before inpainting, we deploy different initialization strategies. We use the source image $S_{rep}$ for replacement directly for the first type of initialization as it has already contained a same-typed subject as the target. However, for Subject Addition, it is challenging to directly generate a context-aware target subject in the designated *Bbox*. Instead, we devise an infill-then-generate schema. With a description of the target subject in the text, GLIGEN-infill can in-paint a subject similar to the target into the desired location in the source image, which is named as GLIGEN initialization merely for Subject Addition. The

infill-then-generate schema enables easy adaptation from the "semi-customized" subject to the target one, as they share similar features. Moreover, the infilled subject tends to establish natural interaction with the context as it is trained with a large amount of image and bounding box pairs (Li et al., 2023b). Meanwhile, we propose COPY as another initialization strategy to preserve the characteristics of the target to the maximum degree. For COPY initialization, we segment the target subject from one of the target subject images $\mathbb{C}_s$, and re-scale it to match the size of the source subject in $S_{rep}$ or the *Bbox* in $S_{add}$. Then we replace the corresponding pixels in $S_{rep}$ or $S_{add}$ with the re-scaled target subject segmentation. We briefly define the initialization step as $x_0^{(1)} = \mathcal{I}(S_{task}, \mathbb{C}_s)$, where $S_{task} \in \{S_{rep}, S_{add}\}$.

**In-painting** Given an input image $x_0^{(i)}$, we segment the source subject from the background with Segment-anything (Kirillov et al., 2023) to obtain the mask $M_s^{(i)}$. Then we dilate $M_s^{(i)}$ to $M_s^{*(i)}$ with morphological transformation and dilation kernel $m$ as $M_s^{*(i)} = \mathcal{D}(x_0^{(i)}, m)$, so that the painted subject can have more space to harmonize with the background. To in-paint the target subject at the dilated mask, with the fine-tuned model $\epsilon_{\theta_s}$, we first encode the input $x_0^{(i)}$ into an intermediate noised latents $x_k^{(i)}$ using DDIM, where $0 < k^{(i)} < T$. Then we denoise $x_k^{(i)}$ conditioned on prompt $Q^*$ with special token for the target subject to obtain $z_t^{(i)}$, where $0 < t < k$. To preserve the background and solely modify the subject, we leverage the dilated segmentation mask $M_s^{*(i)}$ to decide the partial pixels guided by $Q^*$. For the pixels outside $M_s^{*(i)}$, we replace them with the recorded latents in DDIM encoding to reconstruct the background, which can be formulated as $z_t^{*(i)} = (1 - M_s^{*(i)}) * x_t^{(i)} + M_s^{*(i)} * z_t^{(i)}$. Then we can ultimately obtain the denoised image $z_0^{(i)}$ as shown in Figure 3. We formulate the whole in-painting process as $z_0^{(i)} = \mathcal{P}(\epsilon_{\theta_s}, x_0^{(i)}, M_s^{*(i)}, Q^*)$.

**Iterative Generation** According to our observation, the generated subject from the first round may still retain the features from the source subject or cannot interact with the context rationally. Therefore, it motivates us to propose the iterative generation approach as shown in the orange block in Figure 3. In detail, `DreamEditor` $(N)$ will treat the output from the last iteration as the input to the next iteration, i.e. $x_0^{(i+1)} = z_0^{(i)}$, and repeat the in-painting process for all the $N$ iterations. It is worth noting that there is an iterative change of $M_s^{*(i)}$ from round to round. The shape of the segmentation mask will gradually adapt to the target subject so as the generated subject. Briefly, the process is expressed as Algorithm 1.

# 6 Experiments

## 6.1 Experiment Setting

We use stable diffusion version 1.4[1] to fine-tune it with DreamBooth (Ruiz et al., 2023). For all of the 30 subjects involved in DreamBench (Ruiz et al., 2023), we set iteration number $N = 5$ and the mask dilation kernel $m = 20$. The encoding ratio $k_1/T$ is set to be 0.8 for the first iteration and decreases linearly as $k_i/T = k_1/T - i * 0.1$. We leverage the Gligen inpainting pipeline implemented in the official code repository[2] for the initialization of `DreamEditor` for Subject Addition task. Meanwhile, we leverage the text-based Segment-anything implemented at lang-segment-anything[3] to obtain the segmentation mask of the source subject. All the experiments are run on a single A6000 GPU.

## 6.2 Baseline Methods

We compare the proposed `DreamEditor` with the following baselines to show its effectiveness.

**DreamBooth (Ruiz et al., 2023)** We finetune the stable diffusion model with target subject images $\mathbb{C}_s$ and the source image $S_{task}$ to encode the target subject and the source image by special tokens $V^*$ and $Y^*$. Then we prompt the model with *photo of a $V^*$ [class name] in $Y^*$ background.* to obtain the results.

---

[1] https://huggingface.co/CompVis/stable-diffusion-v1-4
[2] https://github.com/gligen/diffusers/tree/gligen
[3] https://github.com/luca-medeiros/lang-segment-anything

---

**Algorithm 1:** `DreamEditor` Algorithm

---

**1** **Input:** A source image $S_{task}$, a target prompt $Q^*$, a fine-tuned stable diffusion model $\epsilon_{\theta_s}$, a set of target subject images $\mathbb{C}_s$, mask dilation kernel $m$ and iteration number $N$.

**2** **Output:** A list of edited images $\mathbb{R}_{sub}$.

**3** $x_0^{(1)} = \mathcal{I}(S_{task}, \mathbb{C}_s)$;

**4** **Function** $\mathcal{P}(\epsilon_{\theta_s}, x_0^{(i)}, M_s^{*(i)}, Q^*)$

**5**    $x_0^{(i)}, x_1^{(i)}, \ldots, x_k^{(i)} = DDIMEncode(\epsilon_{\theta_s}, x_0^{(i)})$;

**6**    $z_k^{(i)} = x_k^{(i)}$;

**7**    **for** $t = k-1, k-2, \ldots, 0$ **do**

**8**       $z_t^{(i)} = DDIMSampler(\epsilon_{\theta_s}, z_{t+1}^{(i)}, Q^*)$;

**9**       $z_t^{*(i)} = (1 - M_s^{*(i)}) * x_t^{(i)} + M_s^{*(i)} * z_t^{(i)}$;

**10**       $z_t^{(i)} = z_t^{*(i)}$;

**11**    **end**

**12** **for** $i = 1, 2, \ldots, N$ **do**

**13**    $M_s^{*(i)} = \mathcal{D}(x_0^{(i)}, m)$;

**14**    $z_0^{(i)} = \mathcal{P}(\epsilon_{\theta_s}, x_0^{(i)}, M_s^{*(i)}, Q^*)$;

**15**    $x_0^{(i+1)} = z_0^{(i)}$;

**16** **end**

**17** $\mathbb{R}_{sub} = \{z_0^{(1)}, z_0^{(2)}, \ldots, z_0^{(N)}\}$;

**18** **Return** $\mathbb{R}_{sub}$

---

| Method | Initialization | Dino-sub↑ | Dino-back↑ | ClipI-sub↑ | ClipI-back↑ | Overall↑ |
|---|---|---|---|---|---|---|
| | | | Subject Replacement | | | |
| DreamBooth | - | 0.718 | 0.481 | 0.867 | 0.744 | 0.608 |
| Customized-DiffEdit | - | 0.619 | **0.878** | 0.834 | **0.915** | 0.790 |
| CopyPaste | COPY | 0.775 | 0.822 | 0.874 | 0.896 | **0.819** |
| PhotoSwap | - | 0.584 | 0.842 | 0.824 | 0.893 | 0.757 |
| CopyHarmonize | COPY | **0.779** | 0.757 | **0.879** | 0.855 | 0.786 |
| `DreamEditor` (1) | COPY | 0.753 | 0.779 | 0.875 | 0.877 | 0.791 |
| `DreamEditor` (5) | COPY | 0.765 | 0.799 | 0.882 | 0.882 | 0.807 |
| `DreamEditor` (1) | - | 0.546 | 0.664 | 0.763 | 0.853 | 0.646 |
| `DreamEditor` (5) | - | 0.564 | 0.667 | 0.77 | 0.855 | 0.655 |
| | | | Subject Addition | | | |
| DreamBooth | - | **0.699** | 0.151 | **0.860** | 0.604 | 0.312 |
| Customized-DiffEdit | GLIGEN | 0.577 | 0.641 | 0.818 | 0.789 | 0.653 |
| CopyPaste | COPY | 0.444 | 0.666 | 0.693 | 0.826 | 0.579 |
| PhotoSwap | GLIGEN | 0.550 | 0.684 | 0.808 | 0.809 | 0.660 |
| CopyHarmonize | COPY | 0.441 | 0.698 | 0.690 | 0.844 | 0.592 |
| `DreamEditor` (1) | COPY | 0.684 | **0.807** | 0.849 | **0.879** | **0.775** |
| `DreamEditor` (5) | COPY | 0.664 | 0.773 | 0.841 | 0.850 | 0.751 |
| `DreamEditor` (1) | GLIGEN | 0.579 | 0.773 | 0.818 | 0.870 | 0.717 |
| `DreamEditor` (5) | GLIGEN | 0.632 | 0.798 | 0.838 | 0.874 | 0.753 |

Table 1: Automatic Evaluation Results of `DreamEditor` and Baselines on Subject Replacement and Addition.

**Customized-DiffEdit (Couairon et al., 2022)** DiffEdit can automatically generate the mask to be edited by contrasting predictions conditioned on source and target prompts, so that it can realize editing without changing the background. We replace the diffusion model in DiffEdit with a fine-tuned one by

| Method | Initialization | Subject↑ | Background↑ | Realistic↑ | Overall↑ |
|---|---|---|---|---|---|
| Subject Replacement | | | | | |
| DreamBooth | - | 0.543 | 0.0 | **0.707** | 0.072 |
| Customized-DiffEdit | - | 0.21 | **0.828** | 0.668 | 0.488 |
| CopyPaste | COPY | **1.0** | 0.148 | 0.123 | 0.263 |
| PhotoSwap | - | 0.15 | 0.773 | 0.663 | 0.425 |
| CopyHarmonize | COPY | **1.0** | 0.552 | 0.147 | 0.433 |
| DreamEditor (1) | COPY | 0.778 | 0.407 | 0.52 | 0.548 |
| DreamEditor (5) | COPY | 0.817 | 0.505 | 0.54 | 0.606 |
| DreamEditor (1) | - | 0.532 | 0.760 | 0.557 | 0.608 |
| DreamEditor (5) | - | 0.630 | 0.800 | 0.582 | **0.664** |
| Subject Addition | | | | | |
| DreamBooth | - | 0.477 | 0.0 | **0.635** | 0.067 |
| Customized-DiffEdit | GLIGEN | 0.288 | 0.302 | 0.252 | 0.280 |
| CopyPaste | COPY | **0.983** | **1.0** | 0.033 | 0.319 |
| PhotoSwap | GLIGEN | 0.21 | 0.562 | 0.305 | 0.33 |
| CopyHarmonize | COPY | **0.983** | **1.0** | 0.295 | **0.662** |
| DreamEditor (1) | COPY | 0.635 | 0.978 | 0.265 | 0.548 |
| DreamEditor (5) | COPY | 0.633 | 0.973 | 0.393 | 0.623 |
| DreamEditor (1) | GLIGEN | 0.287 | 0.99 | 0.427 | 0.495 |
| DreamEditor (5) | GLIGEN | 0.478 | 0.972 | 0.528 | 0.626 |

Table 2: Human Evaluation Results of `DreamEditor` and Baselines on Subject Replacement and Addition.

DreamBooth. Then we modify the source and target prompts to *photo of a [class name]* and *photo of a V* [class name]* correspondingly to enable the generation of the customized subject. For the Subject Addition, we initialize the input the same way as `DreamEditor`.

**Copy-Paste** To have a naive but fundamental baseline, we directly use the output result from COPY initialization. Copy-Paste basically leverages Segment-anything to segment the target subject from one of the provided target subject images set $\mathbb{C}_s$ as "copy" and re-scale it to replace the corresponding pixels in the segmentation box in source image $S_{rep}$ or the labeled bounding box in $S_{add}$.

**PhotoSwap (Gu et al., 2023)** Similar with DreamEditor, PhotoSwap first learns the target concep with DreamBooth, and then swaps it into the target image with attention swap during diffusion process for replacement task. We employ GLIGEN initialization for PhotoSwap in addition task to enable a head-to-head comparison.

**CopyHarmonize** To have a intuitive and straightforward baseline, for subject addition task, we copy-paste for the target subject first, and utilize a Harmonizer (Ke et al., 2022) to contextualize the copied subject in the source image. For subject replacement task, the source subject is segmented out first, and we conduct in-painting on the segmentation mask to fill up the background and repeat the same process with addition task.

### 6.3 Main Results

We run the experiments for `DreamEditor` and baselines on `DreamEditBench`. We present results in Figure 5 and Figure 4. To make a more comprehensive comparison, we evaluate the results with both an automatic matrix and a rigorous human evaluation.

**Automatic Evaluation Results** For both replacement and addition tasks, it is crucial to maintain fidelity to both subject and background. Therefore, we evaluate the fidelity of the generated results with DINO (Caron et al., 2021) and CLIP-I (Radford et al., 2021) scores for both the subject and the back-

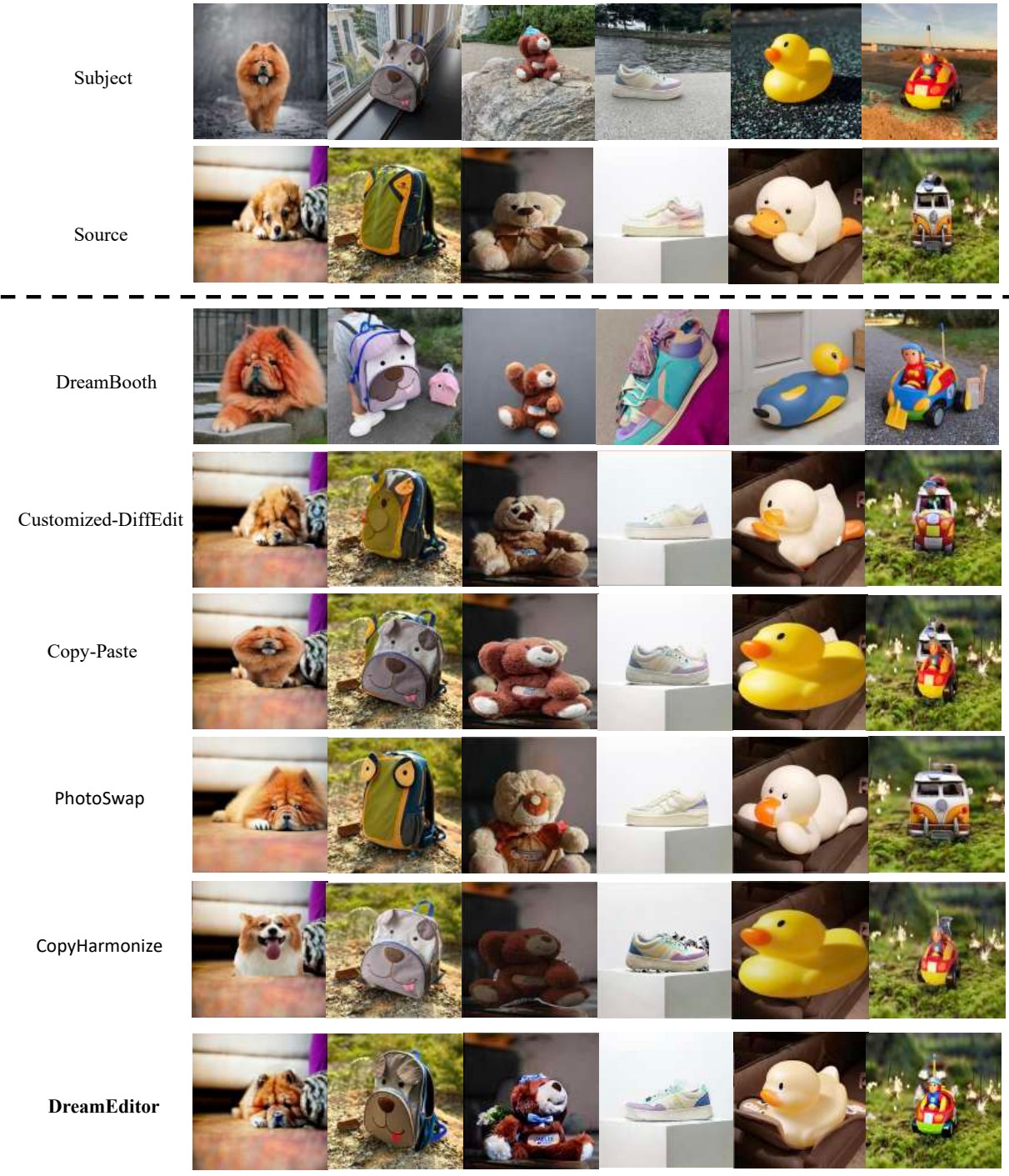

Figure 4: Results on Subject Replacement Task Compared with Baselines

ground. The two matrices measure the average cosine similarity between the generated and real images with ViTS/16 DINO embeddings and CLIP embeddings. We define Dino-sub, Dino-back, ClipI-sub, ClipI-back as the measurements for the subject and background separately. For Dino-sub and ClipI-sub, it deploys all

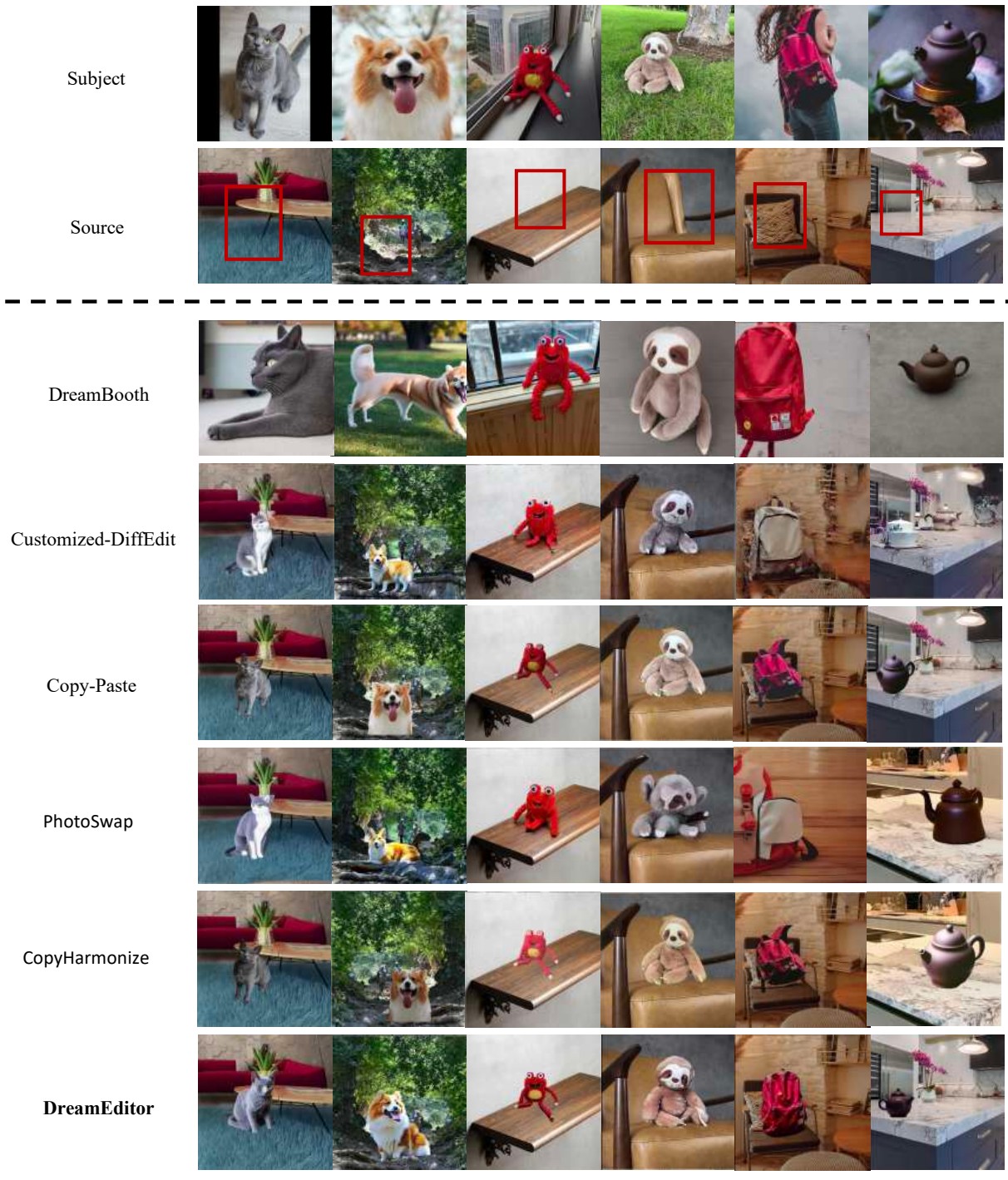

Figure 5: Results on Subject Addition Task Compared with Baselines

the images from the provided target subject images set $\mathbb{C}_s$ as the real images. And for Dino-back and CilpI-back, it leverages the source image $S_{task}$ as the real reference. To disentangle the subject from background or vice versa, the subject is segmented from the background for both the reference and generated images

with off-the-shelf segmentation tool. The segmentation mask or its complementary part will be filled with white for background-oriented or subject-oriented evaluation respectively. The overall score is defined as the average of the geometric mean of the previous four measurements over all the examples. The automatic evaluation results are demonstrated in Table 1. The number in the brackets after DreamEditor refers to the number of iterations.

**Human Evaluation Results** We conduct a human evaluation of the results for `DreamEditBench` to report a more realistic performance. We define three aspects to evaluate the quality of the generated results: 1) Subject Consistency: How well do the generated results preserve the feature of the customized subject in the provided set of images for the target? 2) Background Consistency: How well do the generated results preserve the background information in the source inputs? 3) Realism: Assess the overall realism and naturalness of the generated image. For each of the perspectives, we ask three experts to label the results with scores 0, 0.5, or 1. The detailed evaluation standard is described in Appendix A.2. For methods with several iterations, i.e. `DreamEditor` (5), the expert is asked to decide the best iteration first and evaluate the result for the best iteration. After obtaining the labeling, we average each aspect over all the examples to get the final results as shown in Table 2. The overall score is obtained by calculating the geometric mean of the scores from the three perspectives. Generally, the proposed `DreamEditor` (5) obtains the best overall score among all the methods, surpassing the best baseline Customized-DiffEdit by 0.176 in Subject Replacement. For subject addition task, though CopyHarmonize slightly performs better than `DreamEditor` (5) due to almost full score for Subject and Background consistency, `DreamEditor` (5) outperforms it by 0.233 in Realism score, indicating the higher fidelity of the results generated by `DreamEditor`. The detailed analysis for the comparison of realism is demonstrated in Appendix A.4. Moreover, it is imperative to note that the human evaluation results diverge dramatically from the automatic ones, which reflects the necessity of conducting a rigorous human evaluation for a more fair evaluation among methods.

## 6.4 Result Analysis

As shown in Figure 4 and Figure 5, though with relatively high subject consistency and realism, DreamBooth can hardly preserve the background from the source. Customized-DiffEdit can merely either preserve the background or adapt to the target subject but can barely handle both. It is also difficult for PhotoSwap to adapt to the features of the target subject. CopyPaste although preserves both subject and background consistency, it fails to obtain realistic results in most cases, as it cannot generate context-aware subjects. The realism issue is alleviated in CopyHarmonize baseline, but still yields relatively low realism score. `DreamEditor` with COPY as initialization further mitigates the limitation of CopyHarmonize, leading to more realistic results. It works well when the posture of the segmented subject matches the target context (e.g. monster toy in Figure 5), but fails otherwise (e.g. backpack in Figure 5). `DreamEditor` without initialization for replacement or with GLIGEN initialization for addition balances subject, background, and realism to some extent, thus obtaining the highest overall human evaluation scores.

It can be observed from Table 1, the models initialized with COPY mechanism can achieve better scores in subject-oriented evaluation measurements. While method with weakest background preservation mechanism like DreamBooth performs worst in the background-oriented evaluation matrix, which matches our intuition.

Besides the main results, we conduct a more fine-grained analysis on the sub-division of `DreamEditBench`. As stated in Section 4.3, we divide the collected dataset into easy and hard subsets. We measure the overall results for the two divisions separately for each task as shown in Table 3. For the majority of the cases, the models achieve a better performance on the easy sub-division than the hard one by about 0.1.

In addition, we analyze the effect of the number of iterations in an iterative generation. As shown in Table 4, for Subject Replacement, the Subject Consistency will consistently increase with a larger N. Nevertheless, the Background Consistency and Realism will fluctuate and even decrease with more iterations. Generally, it reflects a trade-off between subject consistency and the other two aspects with the parameter $N$. Thus, if we can pick the best iteration for each example individually, it can achieve the best overall performance.

| Method | Initialization | Subject Replacement | | Subject Addition | |
|---|---|---|---|---|---|
| | | Easy↑ | Hard↑ | Easy↑ | Hard↑ |
| Customized-DiffEdit | -/GLIGEN | **0.511** | 0.418 | **0.303** | 0.250 |
| PhotoSwap | -/GLIGEN | **0.511** | 0.418 | **0.352** | 0.303 |
| CopyHarmonize | -/GLIGEN | **0.511** | 0.418 | **0.685** | 0.632 |
| DreamEditor (1) | COPY | **0.551** | 0.529 | **0.600** | 0.480 |
| DreamEditor (5) | COPY | 0.600 | **0.612** | 0.663 | 0.574 |
| DreamEditor (1) | -/GLIGEN | **0.648** | 0.515 | **0.562** | 0.470 |
| DreamEditor (5) | -/GLIGEN | **0.702** | 0.567 | **0.676** | 0.563 |

Table 3: Comparison of Human Evaluation Results on the Easy and Hard split of `DreamEditBench`.

| Iteration | Subject↑ | Background↑ | Realistic↑ | Overall↑ |
|---|---|---|---|---|
| N=1 | 0.531 | 0.763 | 0.556 | 0.608 |
| N=2 | 0.585 | 0.713 | 0.490 | 0.589 |
| N=3 | 0.601 | 0.693 | 0.456 | 0.574 |
| N=4 | 0.613 | 0.681 | 0.448 | 0.571 |
| N=5 | 0.625 | 0.681 | 0.453 | 0.577 |
| N=**Best** | **0.630** | **0.800** | **0.582** | **0.664** |

Table 4: Human Evaluation Scores with Different Iteration Number for Subject Replacement Task.

## Limitations

`DreamEditor` can either fail or require a large iteration number to adapt when the target subjects differ too much from the source subject. Besides, due to the iterative generation, the error in reconstruction from DDIM inversion will be propagated and leads to a blurry background after $N$ iterations, especially for large $N$. In addition, the performance of `DreamEditor` is highly affected by the segmentation model and GLIGEN in-painting, if the two models fail in the first place, it is very likely that `DreamEditor` will also fail ultimately.

## 7 Conclusion

In this work, we define two novel subject-driven image editing tasks, i.e. Subject Replacement and Subject Addition. To standardize the evaluation of the two proposed tasks, we collect the `DreamEditBench`, containing 440 same-typed subjects and background images for 22 subjects in total. Meanwhile, we devise `DreamEditor` to realize gradual refinement towards target subjects with iterative generation. The systematic human evaluation shows the advantage of our method over the other baselines on the overall performance.

## Broader Impact

The precise control over subjects and backgrounds in image generation, as demonstrated in this research, can contribute to the ongoing efforts to detect and mitigate deepfake content. By improving our understanding of how subjects can interact with their environment, this research can aid in the development of more effective deepfake detection algorithms. In addition, leveraging the proposed method, well-controlled and relatively high-quality fake images can be synthesised automatically, which can be leveraged to train better deepfake detector. By advancing the field of subject-driven image generation, this research indirectly results in the development of countermeasures against malicious uses of deepfake technology. Moreover, it raises awareness among researchers and policymakers about the evolving landscape of AI-generated content, stimulating discussions on potential regulations and safeguards.

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

# A    Appendix

## A.1    Dataset Examples

We demonstrate part of our collected data at Figure 6. The left column is some examples of easy subset and the right column is that for hard subsets.

## A.2    Human Evaluation

To standardize the conduction of a rigorous human evaluation, we stipulate the criteria for each measurement as follows:

**Subject Consistency**  How well do the generated results preserve the feature of the customized subject in the provided set of images for the target subject?

- Score 1: The subject is consistent and accurately represents the intended subject, closely matching the visual characteristics and appearance.

- Score 0.5: The subject partially resembles the intended subject but lacks consistency in some visual attributes, such as facial features, body proportions, or object shapes.

- Score 0: The subject in the generated image bears little resemblance to the intended subject or exhibits significant distortions in visual characteristics and appearance.

**Background Consistency**  How well do the generated results preserve the background information in the provided source inputs?

- Score 1: The background is consistent and seamless, demonstrating a high level of coherence with the given context or scene.

- Score 0.5: The background has some minor inconsistencies or artifacts, but they are not prominent enough to significantly affect the overall coherence of the image.

- Score 0: The background shows significant inconsistencies or artifacts that are visually distracting and do not align with the given context or scene.

**Realism**  Assess the overall realism and naturalness of the generated image.

- Score 1: The generated image is visually convincing and closely resembles a real photograph, exhibiting realistic lighting, shadows, texture details, and overall visual coherence.

- Score 0.5: Score 0.5: The realism of the image is somewhat compromised, showing minor visual flaws or inconsistencies that may raise suspicion but do not strongly detract from its overall appearance.

- Score 0: The generated image appears highly artificial and unrealistic, with noticeable visual flaws, unnatural lighting, or inconsistencies that make it easily identifiable as a generated image.

If the score of an aspect is 0, we use 0.001 as an approximation. We also provide an example for each score with respective to different criteria as shown in Figure 7.

## A.3    Failure Cases

The results are generated by `DreamEditor` with unified parameters for all the examples without subtly tuning for any specific subject. Therefore, there can be various failure cases in different stage of the `DreamEditor` pipeline as shown in Figure 7.

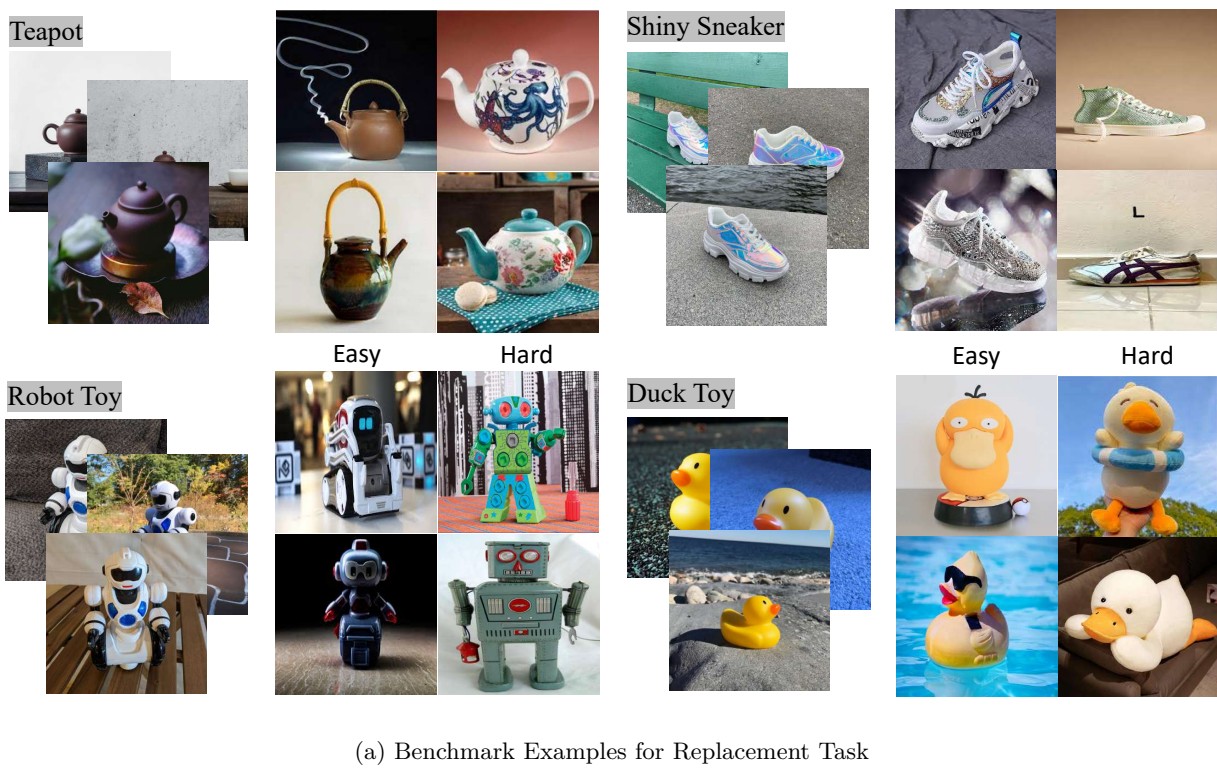

(a) Benchmark Examples for Replacement Task

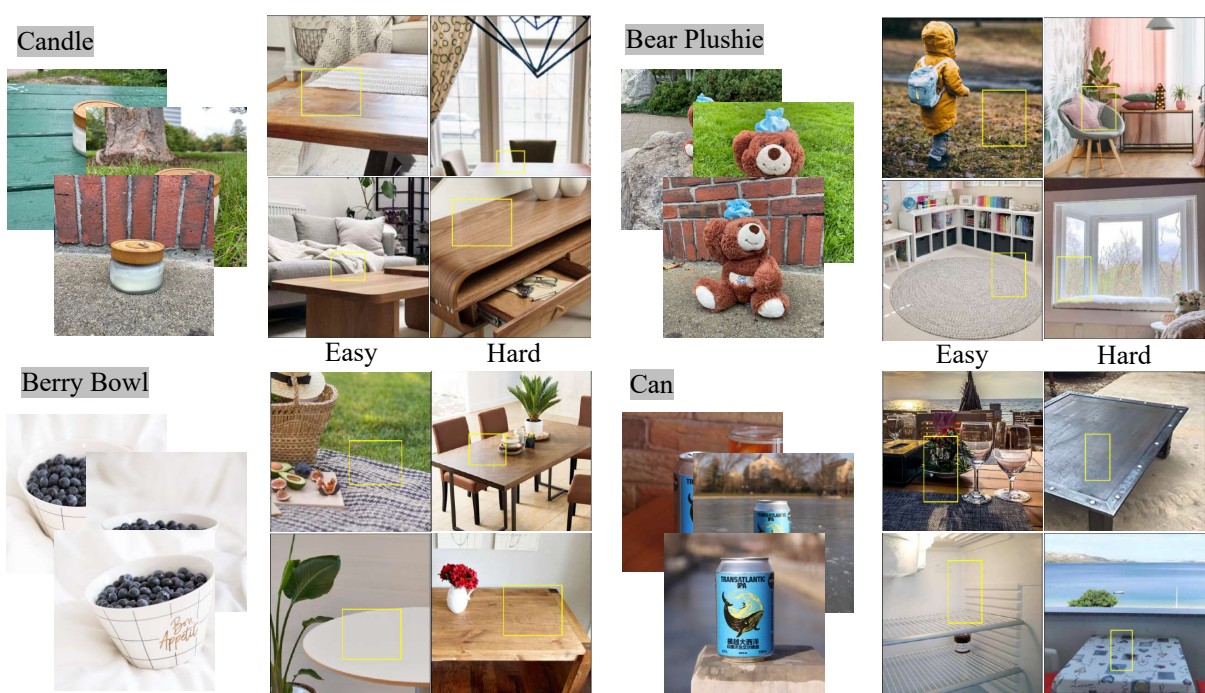

(b) Benchmark Examples for Addition Task

Figure 6: `DreamEditBench` Examples

Subject Consistency    Background Consistency    Realism

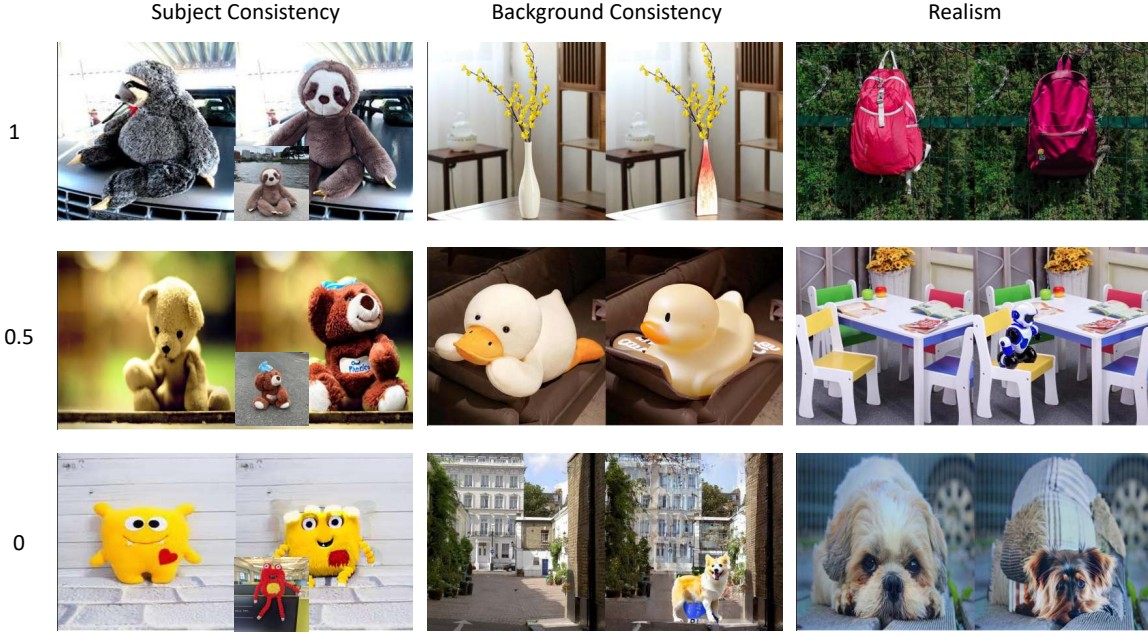

Figure 7: Detailed human evaluation criteria for different measurements.

For instance, in subject replacement task, it is hard for `DreamEditor` to adapt to the target pink backpack from the green source as they share few similar features, and a large iteration number $N$ may be required to transform it completely. Secondly, it will also lead to a failure if the segmentation step fails at the very beginning as shown in the teapot example. The inaccurate segmentation map can result in the shift of the subject position and unnecessary artifact added to the background.

For subject addition task, in addition to all the failures from replacement, the initialization step can play a vital role. It is more likely to fail if the subject generated by GLIGEN differs far from the target subject like the black backpack in the 4th column of Figure 7. Moreover, the subject generated by GLIGEN may not correctly interpret its relationship with the context. The cat should be a real cat sitting at the carpet other than the painting, while the bear plushie should be put on the bay window instead of suspending in the air. These failure cases may be rectified with a more specific parameter tuning for each subject or better segmentation and in-painting model.

### A.4    Comparison with naive baselines

According to the human evaluation results for `DreamEditor` and CopyHarmonize, we conduct deeper analysis on why our method can not outperform the naive CopyHarmonize baseline in subject addition task for the overall score. As we can observe from the left column of Figure 9, when adding the target subject to the source image, CopyHarmonize can achieve exact match with the target subject and preserve the background consistently, as it pastes the segmented subject directly to the background. However, though high scores can CopyHarmonize obtain in subject and background consistency, the copy-and-harmonize pipeline will fail the realism score when the posture or angle of the view play a vital role of making the images look realistic. For instance, as in the first row of the left column in Figure 9, the shoulder straps of a backpack should not stand up when placed in the chair, and it also should not be floating above the chair. In the second row of CopyHarmonize, the angle of the view of the toy car is not consistent with the that of the table from background. Instead, a view of top is expected as generated by `DreamEditor` other than a front view. Moreover, direct copy can lead to obvious artifact when the subject is at the edge of the original image like the flat cut of the body of the dog in the third row of Figure 9. Our `DreamEditor` can generate more

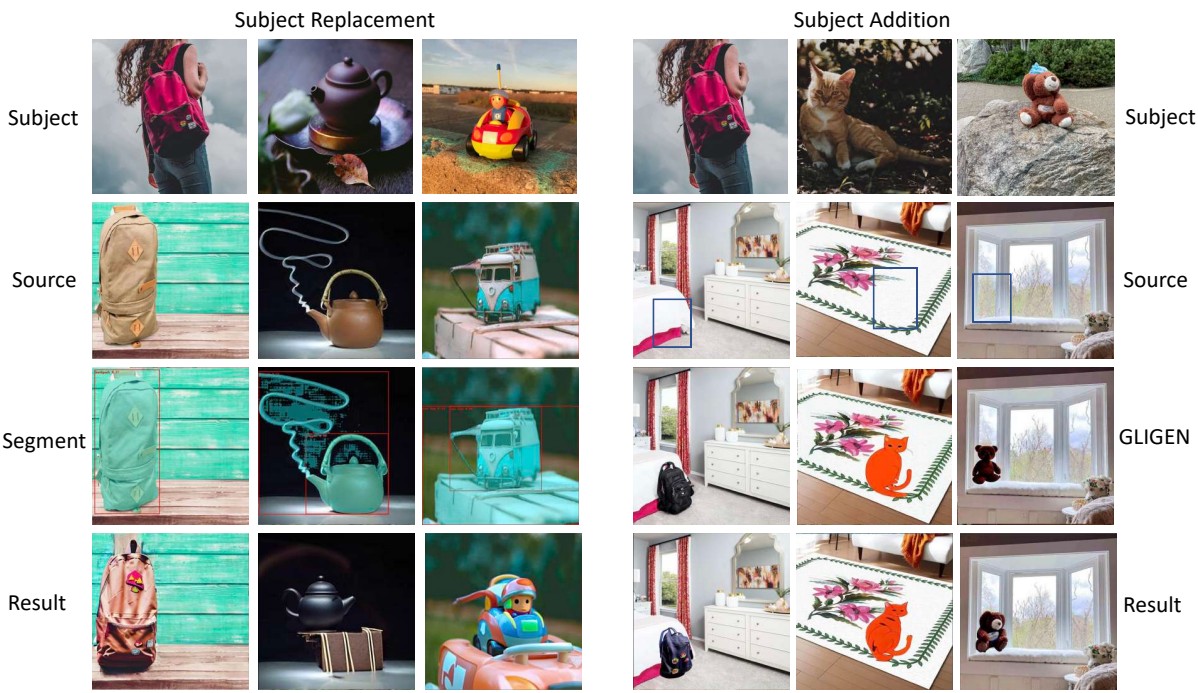

Figure 8: Failure cases of `DreamEditor` in the two proposed tasks.

reasonable and realistic images when it comes to these cases as shown in Figure 9. In subject replacement task, CopyHarmonize will induce patent artifact after inpainting as shown in the right column of Figure 9, which will impair the background consistency and realism of the generated images dramatically.

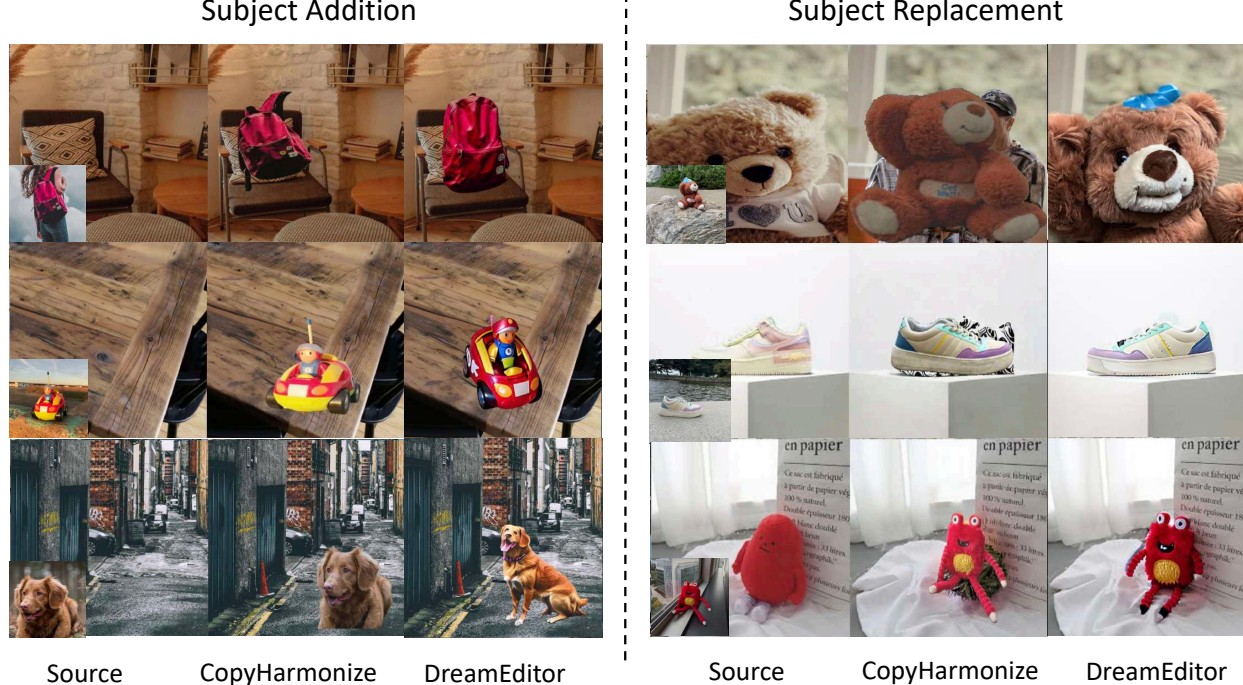

Figure 9: Comparison of `DreamEditor` with CopyHarmonize.

