# OpenReview forum: "DreamEdit: Subject-driven Image Editing"
_TMLR — Accepted by TMLR_

### Review · Reviewer_HtM6 · 2023-07-09

**Summary Of Contributions:**

This work aims at subject-driven image editing and targets two sub-tasks: subject replacement and subject addition. To evaluate the new settings, an evaluation dataset is collected, and several metrics are introduced. An iterative strategy and two initialization schemas are proposed to achieve reasonable performance.

**Audience:**

Yes

**Broader Impact Concerns:**

A broader impact statement should be added to discuss the potential impact on deepfake.

**Claims And Evidence:**

Yes

**Requested Changes:**

Critical adjustments:
- Better baseline comparison.
- Explain the effectiveness of the proposed metrics.

**Strengths And Weaknesses:**

Strengths:

- Subject-driven image editing is a challenging and useful task. This work defines two sub-task settings and proposes to use DreamBooth and an iterative generation scheme to achieve reasonable results.

- The challenges of subject replacement and subject addition are analyzed, and the corresponding evaluation datasets and metrics are introduced. A user study is also constructed to evaluate subject consistency, background consistency, and realism.

- To further improve the subject addition performance, two initialization schemas are introduced: COPY and GLIGEN-infill, which provide a good starting point for subject addition.

- Figure 1 clearly illustrates the two sub-task settings, and the overall paper organization and writing are easy to follow.

Weaknesses:

- The technical contributions are somewhat limited. The iterative generation scheme is not new and has been applied in several previous generation tasks, such as image inpainting “High-Resolution Image Inpainting with Iterative Confidence Feedback and Guided Upsampling, ECCV20”

- A better baseline could be “Inpainting + Copy-Paste + Harmonization,” which is a straightforward workflow for object replacement or addition. It would be good to perform a comparison with such a baseline.

- Regarding the newly introduced evaluation dataset, it only contains 22 subject classes, which may not be enough for a comprehensive evaluation. Collecting at least hundreds of subject classes would be ideal for a more convincing experiment result.

- Regarding the introduced evaluation metrics,  Dino-sub, Dino-back, ClipI-sub, and ClipI-back may be questionable as subject and background pixels are all in the generated image, which may affect the evaluation to each other. For example, when evaluating Dino-sub and ClipI-sub, different backgrounds may cause unpredictable results.

- It is mentioned in the limitation that “DreamEditor can either fail or require a large iteration number” and “leads to a blurry background after N iterations, especially for large N.” The failure cases and analysis could be presented in the appendix.

- It is not clear how to select the best N, and it would be interesting to explore an automatic strategy to choose the best N. For example, we can calculate a metric to do early-stopping.

Minor question: “DreamEditor (5)” in Sec. 6.3, I’m not sure what it means either as it is not explained in the text.

---

### Review · Reviewer_k1Yq · 2023-08-05

**Summary Of Contributions:**

This paper works on subject-driven image editing, including subject replacement and subject addition, by combining DreamBooth, Segment Anything and GLIGEN. The proposed method is evaluated on a newly curated dataset via both quantitive metrics and human evaluation.

**Audience:**

Yes

**Broader Impact Concerns:**

It has a potential mis-use risk to produce fake contents.

**Claims And Evidence:**

Yes

**Requested Changes:**

Regarding the main weakness, I would not request a change since it would be too significant.


We can see that DreamBooth has low background scores. I am curious whether it is possible fine-tune a second model for another special token V2 to learn the background to handle this problem of DreamBooth?

**Strengths And Weaknesses:**

Strengths:

1. The method is evaluated by human evaluation. The observation of the discrepancy between human evaluation and quantitive metrics is beneficial to the community.

2. A new dataset, with moderate size, is curated.

3. The limitations are discussed.

Weaknesses:

1. Technical contribution is limited. The proposed method is a combined application of prior works, i.e., DreamBooth, Segment Anything and GLIGEN. Though it is a recent trend to do research by pipelining, more technical insights are needed.

---

> ### Author Response · Authors · 2023-08-12
>
> We appreciate the reviewer's assessment of our strengths and weaknesses.
>
> **# comment1**: “Technical contribution is limited.”
>
> While our approach leverages existing techniques to build pipelining, our initial experiments show that simple pipelining does not work. The subject generated in one-pass result is  more likely to maintain the characteristics of the source. Therefore, we propose iterative generation with gradual adaptation to the target subject, our novelty in fact lies in the iterative generation, we think the proposed method has two major advantages:
>
> 1) With iterative generation, the dilated segmentation mask from Segment Anything will evolve in-align with the gradual adaptation to the target subject, which boosts the chance of generation with good fidelity (As shown in Table 2, the subject scores of human evaluation increase largely from DreamEditor(1) to DreamEditor(5)).
>
> 2) The stage results of each module are transparent, making it easier to make adjustment and increasing the interpretability of the generated results.
>
> **# comment2:** “DreamBooth baseline”
>
> We have already done this experiment.  As described in section 6.2 in the original submission, we  fine-tuned the diffusion model with both the special token V1 for the target subject and another special token V2 for the background, the reported results are based on this setting. We found that the tuning on background image is hard to maintain faithfulness. The generated backgrounds still contain lots of differences from the original background images.
>
> **Additional experiments / baselines**
>
> In addition, to enable a more comprehensive comparison, we will also include two more baselines, (i.e. "In-painting + Copy-Paste + Harmonization" and  “PHOTOSWAP: Personalized Subject Swapping in Images”) in the revised submission.
>
> We will include a broader impact statement to discuss the potential impact on deepfake in a later version.

---

### Review · Reviewer_4Jvv · 2023-08-23

**Summary Of Contributions:**

This paper introduces an innovative task called subject-driven image editing, encompassing two sub-tasks: subject replacement and subject addition. This task aims to precisely control subject placement, similar to subject-driven image generation, while also managing subject location and pose, akin to conventional image editing. The authors curate a new dataset, named DreamEditBench, dedicated to evaluating this innovative task. Moreover, the authors propose an iterative approach named DreamEditor for this task. DreamEditor employs DDIM for iterative result generation, guided by segmentation masks and text prompts. The method incorporates a customized initialization strategy. Experimental validation on DreamEditBench demonstrates the superiority of the proposed DreamEditor over existing methods in terms of human evaluation results.

**Audience:**

Yes

**Broader Impact Concerns:**

I do not have any concerns about the broader impact of this work.

**Claims And Evidence:**

Yes

**Requested Changes:**

- The equations (1), (2), and (3) lack appropriate punctuation.
- A broader and impartial human evaluation process is recommended.
- Further experimentation involving an ablation study is encouraged.

**Strengths And Weaknesses:**

Strengths:
1. The introduced novel task is very attractive, with the compiled dataset holding promising potential for future research endeavors within this domain.
2. The rationale and details of the method are presented with remarkable clarity, ensuring readers' ease of comprehension. The experiments results are comprehensive, including automatic evaluations, human evaluations and the effect of initialization strategy and different iteration number.

Weakness:
1. While the DreamEditor outperforms existing methods in human evaluation, its results still fall short of achieving the desired level of quality. A noticeable disparity persists between the original subject and the generated counterpart. In many instances, the generated subject fails to seamlessly integrate with the surrounding environment in a plausible manner.
2. The Iteration number is set mannuly for each input image. Regrettably, the model lacks the inherent capability to autonomously ascertain the optimal iteration number for individual images, despite the evident impact of iteration number on the quality of generated outcomes.

---

> ### Author Response · Authors · 2023-08-25
>
> We greatly appreciate the reviewer's insightful comments on our paper.
>
> **Equation Punctuation:** In the revised manuscript, we will ensure that equations (1), (2), and (3) are appropriately punctuated for clarity and consistency.
>
> **Human evaluation:** Our human evaluation is taken for all the baselines together with different settings of DreamEditor in 20 hand-picked source images for each of the 30 subject. It covers subjects with distinct features in color, shape, texture and posture. And the curated datasets include diverse backgrounds with different level of difficulty. We also conduct human evaluation on two more new baselines in the revising version.
>
> **Ablation Study:** We consider the iteration number N of 1 vs 5 is conceived as an ablation study for one-pass generation and the proposed iterative one. Are there any suggestion for other types of ablation study, we will appreciate it!

---

### Author Response · Authors · 2023-09-06
**Manuscript Revision**

We thank the reviewers for their detailed and constructive comments. We replied each concern in the point-to-point responses, and addressed the major requested changes in the submitted revision as follows:
- **More Baseline Comparison**: Add two more baselines to make the experiments more comprehensive: a) PhotoSwap (photoswap: personalized subject swapping in images) b) CopyHarmonize (The “Inpainting + Copy-Paste + Harmonization” mentioned by one of the reviewers)
- **More Rigorous Automatic Evaluation**: Modify the proposed automatic evaluation matrix (i.e. using segmentation tools to separate subject from the background), so that the subject-oriented and background-oriented evaluation matrix are independent of each other.
- **Failure Cases Analysis**: Add a failure cases demonstration and analysis in the appendix.
- **Broader Impact Statement**: Add a broader impact statement about deepfake discussion.
- **Clarity**: Clarify the expression of the model name in the tables.

We value the suggestions provided and welcome any additional recommendations regarding the manuscript.

---

### Author Response · Authors · 2023-11-19
**Final Manuscript**

We thank all the reviewers for the constructive reviews.
We have uploaded the camera ready revision of our work with more result comparison and analysis of DreamEditor and Inpainting + Copy-Paste + Harmonization baseline to show our advantages on hard cases in Appendix A.4.

---

### Decision · Action_Editors · 2023-10-26

**Recommendation:** Accept with minor revision

**Comment:**

Three expert reviewers provided high-quality reviews for this paper. After the revision and the discussions between authors and the reviewers, the ratings are split, with two reviewers leaning to accept and one reviewer leaning to reject.

The primary concerns of the limitation lie in the result quality. In particular, the comparison with the simple baseline "inpaint+paste+harmonization" because it does not require computationally expensive fine-tuning. However, while this simple baseline can preserve the high-resolution details of the input image, it is fundamentally not feasible for more advanced editing beyond simple translation and scaling. For example, in the paper, the authors did show examples of changing the subject's pose in the inserted scene. The revision also provided these baseline results and quantitative comparisons.

Considering all the strengths and weaknesses raised by the reviewers, the AE believes that this paper has sufficient merits to be published at TMLR.

**Audience:**

The TMLR's audience would be interested in learning about this paper.

**Claims And Evidence:**

Yes, the claims and the supporting evidence are convincing.